# The effects of age and sex on cognitive impairment in schizophrenia: Findings from the Consortium on the Genetics of Schizophrenia (COGS) study

Junghee Lee [1,2]*, Michael F. Green[1,2], Keith H. Nuechterlein[1], Neal R. Swerdlow[3], Tiffany A. Greenwood[3], Gerhard S. Hellemann[1,2], Laura C. Lazzeroni[4,5], Gregory A. Light[3,6], Allen D. Radant[7,8], Larry J. Seidman[9,10†], Larry J. Siever[11,12], Jeremy M. Silverman[11,12], Joyce Sprock[3,6], William S. Stone[9,10], Catherine A. Sugar[1,2,13], Debby W. Tsuang[7,8], Ming T. Tsuang[3,14,15], Bruce I. Turetsky[16], Ruben C. Gur[16], Raquel E. Gur[16], David L. Braff[3,6]

1 Department of Psychiatry and Biobehavioral Science, Geffen School of Medicine, University of California Los Angeles, Los Angeles, California, United States of America, 2 VA Greater Los Angeles Healthcare System, Los Angeles, California, United States of America, 3 Department of Psychiatry, University of California San Diego, La Jolla, California, United States of America, 4 Department of Psychiatry and Behavioral Science, Stanford University, Stanford, California, United States of America, 5 Department of Veterans Affairs Health Care System, Palo Alto, California, United States of America, 6 VISN22, Mental Illness Research, Education and Clinical Center (MIRECC), VA San Diego Healthcare System, San Diego, California, United States of America, 7 Department of Psychiatry and Behavioral Science, University of Washington, Seattle, Washington, United States of America, 8 VA Puget Sound Healthcare System, Seattle, Washington, United States of America, 9 Department of Psychiatry, Harvard Medical School, Boston, Massachusetts, United States of America, 10 Massachusetts Mental Health Center Public Psychiatry Devision of the Beth Israel Deaconess Medical Center, Boston, Massachusetts, United States of America, 11 Department of Psychiatry, The Mount Sinai School of Medicine, New York, New York, United States of America, 12 James J. Peters VA Medical Center, New York, New York, United States of America, 13 Department of Biostatistics, University of California Los Angeles, Los Angeles, California, United States of America, 14 Institute for Genomic Medicine, University of California, San Diego, California, United States of America, 15 Harvard Institute of Psychiatry Epidemiology and Genetics, Boston, Massachusetts, United States of America, 16 Department of Psychiatry, University of Pennsylvania, Philadelphia, Pennsylvania, United States of America

† Deceased.
* jungheelee@uabmc.edu

## Abstract

Recently emerging evidence indicates accelerated age-related changes in the structure and function of the brain in schizophrenia, raising a question about its potential consequences on cognitive function. Using a large sample of schizophrenia patients and controls and a battery of tasks across multiple cognitive domains, we examined whether patients show accelerated age-related decline in cognition and whether an age-related effect differ between females and males. We utilized data of 1,415 schizophrenia patients and 1,062 healthy community collected by the second phase of the Consortium on the Genetics of Schizophrenia (COGS-2). A battery of cognitive tasks included the Letter-Number Span Task, two forms of the Continuous Performance Test, the California Verbal Learning Test, Second Edition, the Penn Emotion Identification Test and the Penn Facial Memory Test. The effect of age and gender on cognitive performance was examined with a general linear model. We observed age-related changes on most cognitive measures, which was similar between

**Data Availability Statement:** Data are contained within Supporting Information File.

**Funding:** This study was supported by grants R01-MH065571, R01-MH065588, R01-MH065562,

R01-MH065707, R01-MH065554, R01-MH065578, R01-MH065558, R01-MH86135, and K01-MH087889 from the National Institute of Mental Health. Other than providing support, the National Institute of Mental Health does not have any further role in this manuscript.

**Competing interests:** Dr. Green has been a paid consultant for Biogen, Click Therapeutics, Lundbeck, and Roche. He is a member of the Scientific Advisory Board of Cadent. Dr. Light has been a paid consultant for Astellas Pharma, Inc, Hepatares Therapeutics, NeuroSig, and Taketa Phamaceutical Company, Ltd, and received grants from Boerhinger-Ingelheim. Dr. Nuechterlein received grants from Janssen Pharmaceutica, Posit Science Corporation, and has been a paid consultant for Janssen Pharmaceutica, Astellas Pharma, Inc, Biogen, Inc., Genetech, Inc, MedinCell, Otsuka Pharmaceutical Co, Ltd, Takeda Pharmaceutical Company, Ltd, and Teva Pharmaceutical Industries, Ltd. No other disclosure is reported. It should be noted that this does not alter our adherence to PLOS ONE policies on sharing data and materials.

males and females. Compared to controls, patients showed greater deterioration in performance on attention/vigilance and greater slowness of processing social information with increasing age. However, controls showed greater age-related changes in working memory and verbal memory compared to patients. Age-related changes ($\eta^2_p$ of 0.001 to .008) were much smaller than between-group differences ($\eta^2_p$ of 0.005 to .037). This study found that patients showed continued decline of cognition on some domains but stable impairment or even less decline on other domains with increasing age. These findings indicate that age-related changes in cognition in schizophrenia are subtle and not uniform across multiple cognitive domains.

## Introduction

While schizophrenia is considered a neurodevelopmental disorder with fixed early deficits, emerging evidence also supports the idea that schizophrenia patients may show both an early brain dysfunction along with progressive neural and/or behavioral deterioration over the course of the illness. For example, schizophrenia patients showed greater age-related reduction in the brain gray matter volumes than controls after early adulthood [1, 2]. This pattern was disproportionately greater in the clinically important frontal and temporal regions [2]. Schizophrenia patients also showed faster age-related deterioration in white matter integrity, emerging after age 35, and this deterioration was focused in the anterior corpus callosum and the temporal aspects of the superior longitudinal fasciculus, also areas implicated in schizophrenia [2–4]. Interestingly, age-related deterioration in both gray and white matter appears to be faster in males than females [2], consistent with the idea that schizophrenia pathology may show sexual dimorphism with males showing greater impairment across multiple neurobiological and clinical domains [5, 6]. Patients also showed greater age-related decline in functional connectivity in several neural networks including the frontal-parietal network and cingulo-opercular network compared with healthy control subjects (HCS) [7]. Thus, existing evidence suggests that schizophrenia patients show faster age-related decline in the structure and function of the brain compared to healthy controls.

The evidence for accelerated aging in the structure and function of the brain in schizophrenia raises questions about its potential consequences for cognitive function. Longitudinal studies with recent-onset psychotic patients shows that impaired cognitive performance was largely stable up to 20 years past onset [8–11]. However, because most studies did not assess healthy controls at follow-up, the stable pattern of cognitive performance in patients does not necessarily rule out that possibility that patients show greater age-related deterioration. Further, most studies on cognitive changes over the course of illness did not assess multiple cognitive domains nor did they examine whether female and male patients show different patterns of performance over time. In addition, most studies have examined only non-social cognition. A recent finding of stable social cognitive performance over 5 years in recent-onset schizophrenia [12] suggests, but did not demonstrate, that age-related changes in cognition in schizophrenia may be similar across several cognitive domains.

To examine these questions regarding stability versus progression of cognitive deficits with age in schizophrenia patients versus healthy controls, this study utilized a large between-subjects data set from a large case-control study of endophenotypes collected by the second phase of the Consortium on the Genetics of Schizophrenia (COGS-2) [13]. Specifically, this study included a large battery of cognitive tasks tapping into multiple cognitive domains to examine

1) whether schizophrenia patients show age-related changes in cognition that differ from those in healthy controls, and 2) if age-related effects differ between males and females.

## Materials and methods

Participants included 1,415 patients diagnosed with schizophrenia or schizoaffective disorder, depressed type, via a Structured Clinical Interview for DSM-IV (SCID) and 1,062 healthy community controls from the COGS-2 study. The NIH-funded COGS-2 included data collection from five sites: University of California at Los Angeles (UCLA), University of California at San Diego (UCSD), Mount Sinai School of Medicine (MSSM), University of Pennsylvania (PENN), and University of Washington (UW). The COGS-2 study also included Stanford University as a non-data-collecting site. Details of the recruitment, the selection criteria for participants and clinical assessments are available in the Supplement and elsewhere [13]. We decided to recruit patients with schizophrenia or schizoaffective disorder, depressed type, for the following reasons. First, findings from genetic studies suggest that schizoaffective disorder, depressed type, is more genetically similar to schizophrenia compared to schizoaffective disorder, bipolar type [14, 15]. Second, schizoaffective disorder, depressed type, showed illness course similar to that of schizophrenia, compared to schizoaffective disorder, bipolar type [16, 17]. The local Institutional Review Boards of each site approved the study, and all participants provided informed consent and were compensated for their participation. For both patients and controls, the Global Assessment of Function Scale (GAF) [18] was administered to characterize samples. Additional clinical assessments for patients included a modified version of the Scale for the Assessment of Negative Symptoms (SANS) [19] and the Scale for the Assessment of Positive Symptoms (SAPS) [20].

### Assessments on cognition

The following cognitive tasks from the COGS-2 study were included in this analysis: the Letter-Number Span Task (LNS) [21, 22] for working memory; two forms of the Continuous Performance Test (CPT) [23–25] for attention/vigilance; the California Verbal Learning Test, Second Edition (CVLT-II) [26, 27] for verbal memory; and the Penn Emotion Identification Test (PEIT) and PENN Facial Memory Test (PEMT) of the Penn Computerized Neurocognitive Battery (CNB) [28, 29] for facial affect recognition and facial memory, respectively. As details of each paradigm are described in the references cited, brief descriptions are provided below.

The LNS [30] employed a set of intermixed letters and digits that experimenters verbally presented at a rate of one per second and consisted of two conditions, the "Forward" and "Reorder" conditions. In the Forward condition, participants were asked to repeat the letters and numbers in the same order as they were presented. In the Reorder condition, participants were asked to repeat the digits first in ascending order and then letters in alphabetical order. The primary measure for each condition was the total number of correctly recalled sequences (i.e., maximum score for each condition = 21).

The two forms of CPT involved computerized versions of the Degraded Stimulus CPT (DS-CPT) and the CPT-Identical Pairs (CPT-IP) [23]. The DS-CPT employed a quasi-random series of blurred single digits that were presented for 29-ms each at a rate of one digit per second. Participants were asked to detect each target "0". The CPT-IP employed a series of digits presented for 50 ms in a quasi-random sequence at a rate of one digit per second and had two conditions: a 3-digit condition and a 4-digit condition. Participants were asked to respond whenever the same stimulus occurred twice in a row. For both the DS-CPT and CPT-IP, the

primary measure was d prime (d'), a signal/noise discrimination index reflecting attention and vigilance from signal detection theory [31, 32].

The CVLT-II employed a 16-item word list that was presented verbally over five learning trials, followed by a single presentation of a second list of 16-item words [26]. Participants were asked to recall as many words as possible from the first list immediately after each list presentation, after the second list was presented, and then after a 20-minute delay. The primary measure was Trials 1–5 Free Recall Correct, Short-Delay Free Recall, and Long-Delay Free Recall.

The PENN Emotion Recognition Test (PEIT) of the Penn computerized neurocognitive battery (CNB) [28, 29] was employed as a measure of emotion identification. The PEIT employed 40 facial stimuli expressing one of 4 emotions (happy, sad, anger, fear) or neutral facial expressions that were presented one at a time. Participants were asked to identify the emotion displayed. The primary measures were accuracy and reaction time. The PENN Face Memory Test (PEMT) [28, 29] was employed to assess episodic memory for faces. In the first part of the task, the participants were shown a series of 20 faces one at a time and asked to memorize them. During the recognition part, the participants were shown 40 faces, 20 of which were the faces they were shown, and the other 20 were distractors. For each face stimulus, participants were asked to indicate whether they have seen the face before. The primary measures were accuracy and reaction time.

## Statistical analysis

For demographic and clinical characteristics, differences by diagnostic group and gender were examined with ANOVA for continuous variables and chi-square tests for categorical variables. The effect of age and gender on cognitive performance in schizophrenia was examined with a general linear model. Specifically, for each dependent measure of the cognitive tasks, a general linear model was conducted, with age, gender and group as fixed factors. Age was included as a continuous variable and gender and group were included as categorical variables. In addition to the linear effect of age, we also examined whether age has a non-linear effect on cognition by adding the quadratic effect of age in the model. The addition of the non-linear effect of age did not improve the model fit, and we report findings from a general linear model with the linear effect of age below. Effect sizes are provided for any significant effects (i.e., partial eta square [$\eta^2_p$]s: $\eta^2_p$ of 0.01 represents a small; $\eta^2_p$ of 0.06 represents a medium; and $\eta^2_p$ of 0.14 represents a large effect size [33].

## Results

### Demographic and clinical characteristics of the participants

Table 1 presents demographic and clinical characteristics of the participants. For age, we observed a significant effect of group ($F_{(1,2473)} = 252.88$, $p < .001$, $\eta^2_p = .093$) and a significant group by sex interaction ($F_{(1,2473)} = 14.52$, $p < .001$, $\eta^2_p = .006$). Overall, controls were younger than patients ($p < .001$). Females were younger than males in the control group ($p < .001$), but not in the patient group ($p = .055$). For personal education, we observed a significant effect of group ($F_{(1,2472)} = 745.44$, $p < .001$, $\eta^2_p = .232$), a significant effect of sex ($F_{(1,2472)} = 13.35$, $p < .001$, $\eta^2_p = .005$) and a significant group by sex interaction ($F_{(1,2472)} = 7.79$, $p < .01$, $\eta^2_p = .003$). Females had higher personal education levels than males in the control group ($p < .001$), but not in the patient group ($p = .52$). For parental education, we found a significant effect of group ($F_{(1,2269)} = 146.62$, $p < .001$, $\eta^2_p = .061$) and a significant group by sex interaction ($F_{(1,2269)} = 6.32$, $p < .05$, $\eta^2_p = .003$). Males had higher levels of parental education than females in the patient group ($p < .05$), but not in the control group ($p = .14$). For GAF in the

**Table 1. Demographic and clinical characteristics of participants.**

|  | Patients | | Controls | |
|---|---|---|---|---|
|  | Female (n = 436) | Male (n = 979) | Female (n = 540) | Male (n = 522) |
| Age | 47.2 (10.5) | 45.9 (11.2) | 37.3 (13.4) | 39.8 (12.7) |
| Personal Edu. | 12.5 (2.2) | 12.5 (2.0) | 15.2 (2.1) | 14.7 (2.2) |
| Parental Edu. | 12.1 (3.1) | 12.5 (2.9) | 14.1 (2.9) | 13.8 (3.1) |
| % Past Sub Dis |  |  |  |  |
| GAF | 44.7 (8.7) | 43.1 (7.8) | 87.5 (7.6) | 85.2 (8.5) |
| Age of Onset | 23.2 (3.1) | 21.8 (6.2) |  |  |
| Global_SANS | 10.8 (6.0) | 11.1 (5.5) |  |  |
| Global_SAPS | 6.7 (4.1) | 7.0 (3.9( |  |  |

** GAF: GAF of the last month

past month, we observed a significant effect of group ($F_{(1,2454)}$ = 15438.32, p < .001, $\eta^2_p$ = .863) and a significant effect of sex ($F_{(1,2454)}$ = 33.95, p < .001, $\eta^2_p$ = .014). Patients had lower GAF than controls (p < .001), and across both groups, females had higher GAF than males (p < .001). Within the patient group, males reported younger onset of psychotic symptoms than females ($F_{(1,1399)}$ = 12.79, p < .001, $\eta^2_p$ = .009). Male and female patients did not differ on positive or negative symptoms (p = .15 and p = .46, respectively).

## The effect of age and sex on cognitive performance

For LNS Forward (Fig 1a), we observed a significant effect of age ($F_{(1,2415)}$ = 74.33, p < .001, $\eta^2_p$ = .019), a significant effect of group ($F_{(1,2415)}$ = 55.07, p < .001, $\eta^2_p$ = .022), and a significant group by age interaction ($F_{(1,2415)}$ = 5.81, p < .05, $\eta^2_p$ = .002). No other effect was significant. Patients showed poorer performance than controls. While both groups showed the decline in performance with increasing age, this decline was steeper in the control group. For LNS Reorder (Fig 1b), we observed a significant effect of age ($F_{(1,2410)}$ = 55.89, p < .001, $\eta^2_p$ = .022) and a significant effect of group ($F_{(1,2410)}$ = 46.82, p < .001, $\eta^2_p$ = .019), but no other effect was significant. Patients showed poorer performance than controls and both groups showed the decline in performance with increasing age.

For DS-CPT (Fig 1c), we found a significant effect of age ($F_{(1,2184)}$ = 70.73, p < .001, $\eta^2_p$ = .031) and a significant effect of group ($F_{(1,2187)}$ = 12.46, p < .001, $\eta^2_p$ = .005). No other effect was significant. In both groups, performance declined with increasing age. Patients performed poorer than controls. For the CPT-IP 3-digit condition (Fig 1d), we found a significant effect of age ($F_{(1,2241)}$ = 32.12, p < .001, $\eta^2_p$ = .014), a significant effect of group ($F_{(1,2241)}$ = 18.88, p < .001, $\eta^2_p$ = .008) and a significant group by age effect ($F_{(1,2241)}$ = 4.46, p < .05, $\eta^2_p$ = .001). Patients showed poorer performance than controls. While both groups showed the decline in performance with increasing age, this effect was greater in the patient group than the control group. For CPT-IP with a 4 digit condition (Fig 1e), we found a significant effect of age ($F_{(1,2193)}$ = 38.57, p < .001, $\eta^2_p$ = .017) and a significant effect of group ($F_{(1,2193)}$ = 34.77, p < .001, $\eta^2_p$ = .015). Both groups showed the decline in performance with increasing age and patients showed poorer performance than controls.

For CVLT-II Trials 1–5 Free Recall Correct (Fig 1f), an effect of age and an effect of group were significant ($F_{(1,2382)}$ = 129.10, p < .001, $\eta^2_p$ = .051; and $F_{(1,2382)}$ = 93.62, p < .001, $\eta^2_p$ = .037, respectively). In both groups, performance declined with increasing age. Patients showed poorer performance. For CVLT-II short-delay free recall (Fig 1g), we found a significant effect

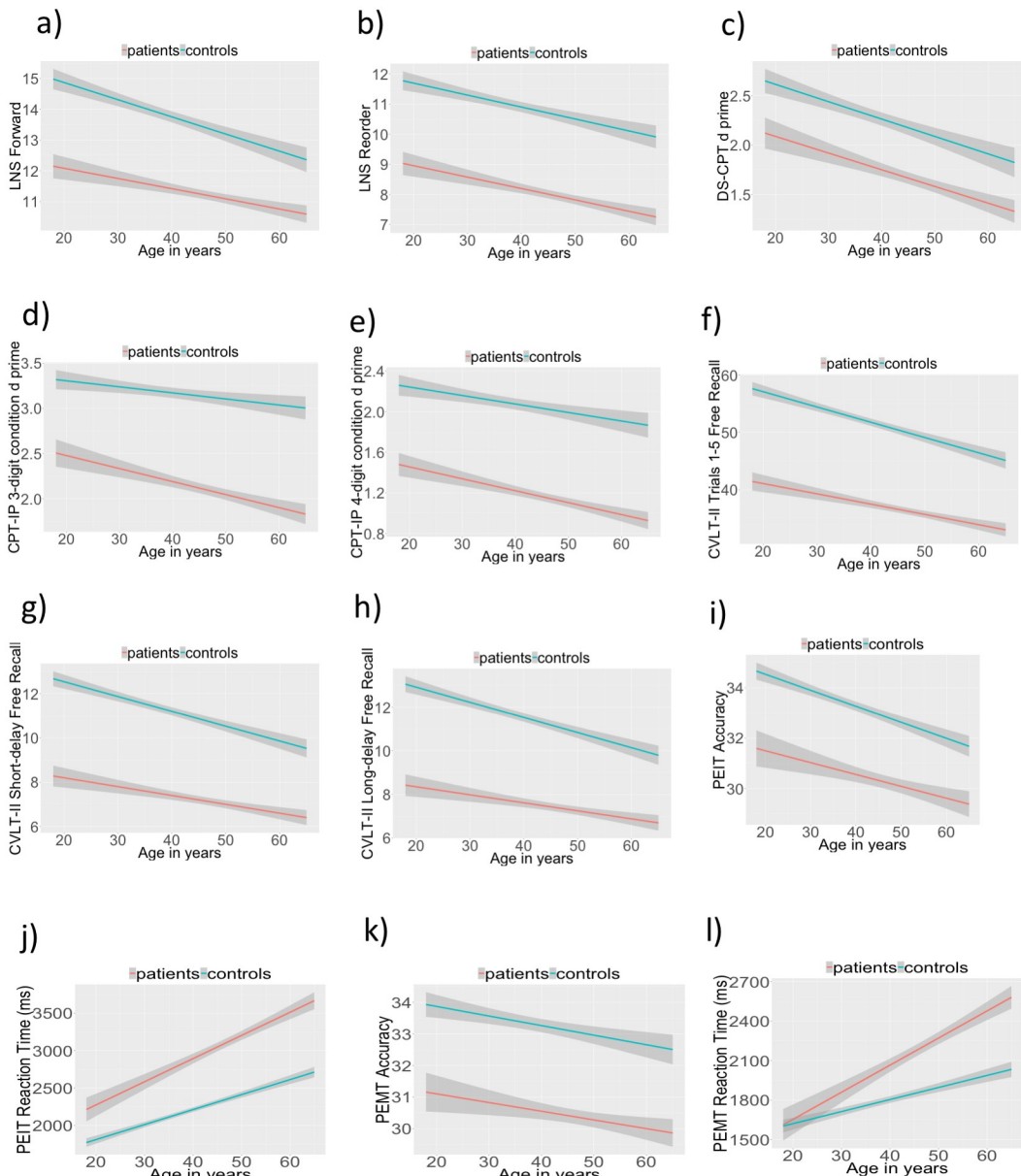

**Fig 1. Cognitive performance in patients (red line) and controls (blue line).** Lines show fitted value of cognitive performance with increasing age, and shaded areas correspond to 95% confidence interval.

of age ($F_{(1,2382)}$ = 86.43, p < .001, $\eta^2_p$ = .035), a significant effect of group ($F_{(1,2382)}$ = 82.64, p < .001, $\eta^2_p$ = .033), and a significant group by age interaction ($F_{(1,2382)}$ = 3.97, p < .05, $\eta^2_p$ = .001). Patients showed poorer performance than controls. Both groups showed the decline in performance with increasing age, and this aging effect was larger in the control group. Similarly, for CVLT-II long-delay free recall (Fig 1h), we found a significant effect of age ($F_{(1,2382)}$ = 76.75, p < .001, $\eta^2_p$ = .031), a significant effect of group ($F_{(1,2382)}$ = 85.44, p < .001, $\eta^2_p$ = .034), and a significant group by age interaction ($F_{(1,2382)}$ = 5.34, p < .05, $\eta^2_p$ = .002). Patients again showed poorer performance than controls. Both groups showed the decline in performance with increasing age, with the steeper decline in the control group.

For PEIT accuracy (Fig 1i), we found a significant effect of age ($F_{(1,2121)}$ = 53.69, p < .001, $\eta^2_p$ = .024) and a significant effect of group ($F_{(1,2121)}$ = 22.67, p < .001, $\eta^2_p$ = .010). Both groups showed poorer performance with increasing age, and patients showed poorer performance than controls. For PEIT reaction time (Fig 1j), we found a significant effect of age ($F_{(1,2121)}$ = 262.49, p < .001, $\eta^2_p$ = .110) and a significant group by age interaction ($F_{(1,2121)}$ = 12.71, p < .001, $\eta^2_p$ = .005). Both groups showed slower reaction time with increasing age, and this effect was larger in the patient group. For accuracy of the PEMT (Fig 1k), we observed a significant effect of age ($F_{(1,2157)}$ = 16.44, p < .001, $\eta^2_p$ = .007) and a significant effect of group ($F_{(1,2157)}$ = 17.78, p < .001, $\eta^2_p$ = .008). Patients showed poorer performance than controls. In both groups performance declined with increasing age. For reaction time in the PEMT task (Fig 1l), we found a significant effect of age ($F_{(1,2157)}$ = 144.02, p < .001, $\eta^2_p$ = .062) and a significant effect age by group interaction ($F_{(1,2157)}$ = 18.91, p < .001, $\eta^2_p$ = .008). Both groups showed slower reaction time with increasing age and this effect was greater in the patient group.

## Discussion

In this study, using a large sample of chronic, medicated outpatients and controls of COGS2 study, we examined whether schizophrenia patients and healthy controls show differential age-related differences in cognition and whether age-related changes differ between male and female participants. For most cognitive measures, we observed selective age-related differences between controls and patients that did not differ between male and female participants. Specifically, compared to controls, schizophrenia patients showed greater deteriorated performance with increasing age on attention/vigilance and greater slowness for processing social information. Compared to patients, controls showed greater aging effect on working memory and verbal memory. Further, age-related changes in these cognitive measures were subtle as evidenced by small effect sizes ($\eta^2_p$ of 0.001 to .008), and much smaller than the prominent between-group differences (i.e., patients showing poorer performance than controls; $\eta^2_p$ of 0.005 to .037). This rather subtle effect of accelerated aging on cognition could be detected because of the large sample size.

Previous longitudinal studies with recent-onset psychotic patients showed stable cognitive performance after the onset of illness [8–11]. However, because most studies had small samples and did not assess healthy controls at follow-up, it remained unclear whether schizophrenia patients showed age-related decline in cognition over time and, if so, whether this decline differs from that of controls. We observed greater age-related cognitive deterioration of patients on some cognitive domains, suggesting a subtle progressive degenerative process. However, on other cognitive domains, we observed smaller age-related effects in schizophrenia compared to controls, but there was still substantial impairment in schizophrenia patients across different ages. This pattern suggests relatively stable levels of cognitive impairment after the onset of illness. Given the critical role of cognitive deficits for functional outcome in schizophrenia [34], our finding of continued decline of cognition along with substantial impairment during adulthood suggests that pharmacological and psychosocial interventions to improve cognitive deficits could benefit schizophrenia patients in all phases of illness.

These findings raise the question of which factors contribute to accelerated cognitive aging in schizophrenia when it is seen. One possibility is that the cognitive decline could be due to accelerated brain aging of schizophrenia patients [1, 2, 7, 35]. Faster brain aging based on neuroimaging in schizophrenia was observed across multiple brain regions, which is consistent with the observation from the current study that patients showed accelerated cognitive aging on attention/vigilance and information processing speed. However, accelerated cognitive aging of patients seen in this study was smaller than one might expect based on the effect sizes

of the age-related changes at the neural level. We also did not find greater age-related changes on working memory and verbal memory, even though other studies have found that age-related reduction in the brain gray matter volume in schizophrenia was greater in the frontal and temporal regions [2]. It remains to be determined whether other personal factors (e.g., educational level, cognitive reserve) may play a role of protecting performance from age-related changes in schizophrenia.

The changes for social cognition with age in schizophrenia were observed only in reaction time, not accuracy. In studies with healthy adults, declines with aging appear first in processing speed-based measures, followed by accuracy-based measures [36, 37]. Given that age-related decline was observed with accuracy measures on non-social cognitive tasks in this study, one may wonder whether non-social cognition is more vulnerable to age-related changes than social cognition in schizophrenia. However, we did not measure reaction time for non-social cognitive measures, so we cannot directly test this possibility.

This study had several limitations. It employed a cross-sectional design, and it was not possible to determine whether the same pattern of findings would be seen with a longitudinal study. This study included performance-based cognitive measures, and it remains to be determined whether a similar pattern of accelerated aging would be seen in neural measures of cognition. While this study included individuals over a wide range of ages, we did not include individuals older than 60. Thus, we do not know if a similar level of age-related cognitive decline would be present among patients older than 60 years old.

In summary, with a large sample of schizophrenia patients and controls, this study examined the stability versus progression of cognitive deficits in schizophrenia. Schizophrenia patients showed small, but significantly greater, age-related cognitive decline on attention/vigilance and social cognition (i.e., reaction time) but less age-related changes on working memory and verbal memory, compared to controls. Further, this age-related change in cognitive performance in schizophrenia was much smaller than the substantial between-group difference and was similar across both male and female patients. The current findings indicate that age-related changes in cognition in schizophrenia are subtle and are not uniform across multiple cognitive domains.

## Supporting information

**S1 Data.**
(CSV)

**S1 Methods.**
(DOCX)

## Acknowledgments

Dr. Larry J. Seidman passed away before the submission of the final version of this manuscript. Dr. Junghee Lee accepts responsibility for the integrity and validity of the data collected and analyzed.

## Author Contributions

**Conceptualization:** Junghee Lee, Michael F. Green, Keith H. Nuechterlein, David L. Braff.

**Data curation:** Joyce Sprock.

**Formal analysis:** Junghee Lee, Michael F. Green, Gerhard S. Hellemann, Catherine A. Sugar.

**Funding acquisition:** Michael F. Green, Keith H. Nuechterlein, Neal R. Swerdlow, Laura C. Lazzeroni, Gregory A. Light, Allen D. Radant, Larry J. Seidman, Larry J. Siever, Jeremy M. Silverman, Joyce Sprock, William S. Stone, Catherine A. Sugar, Debby W. Tsuang, Ming T. Tsuang, Bruce I. Turetsky, Ruben C. Gur, Raquel E. Gur, David L. Braff.

**Investigation:** Michael F. Green.

**Writing – original draft:** Junghee Lee, Michael F. Green, Keith H. Nuechterlein, Gerhard S. Hellemann.

**Writing – review & editing:** Junghee Lee, Michael F. Green, Keith H. Nuechterlein, Neal R. Swerdlow, Tiffany A. Greenwood, Gerhard S. Hellemann, Laura C. Lazzeroni, Gregory A. Light, Allen D. Radant, Larry J. Siever, Jeremy M. Silverman, Joyce Sprock, William S. Stone, Catherine A. Sugar, Debby W. Tsuang, Ming T. Tsuang, Bruce I. Turetsky, Ruben C. Gur, Raquel E. Gur, David L. Braff.

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
