## [Decision Letter · Decision Letter 0]

18 Feb 2020

PONE-D-20-00548

The Effects of Age and Sex on Cognitive Impairment in Schizophrenia: Findings from the Consortium on the Genetics of Schizophrenia (COGS) Study

PLOS ONE

Dear Dr. Lee,

Thank you for submitting your manuscript to PLOS ONE. After careful consideration, we feel that it has merit but does not fully meet PLOS ONE’s publication criteria as it currently stands. Therefore, we invite you to submit a revised version of the manuscript that addresses the points raised during the review process.

The reviewers addressed some minor concerns about your manuscript. Please revise your manuscript carefully.

We would appreciate receiving your revised manuscript by Apr 03 2020 11:59PM. To enhance the reproducibility of your results, we recommend that if applicable you deposit your laboratory protocols in protocols.io, where a protocol can be assigned its own identifier (DOI) such that it can be cited independently in the future. For instructions see: http://journals.plos.org/plosone/s/submission-guidelines#loc-laboratory-protocols

We look forward to receiving your revised manuscript.

Kind regards,

Kenji Hashimoto, PhD

Academic Editor

PLOS ONE

Journal Requirements:

"Dr. Green has been a paid consultant for Biogen, Click Therapeutics, Lundbeck, and Roche. He is a member of the Scientific Advisory Board of Cadent. Dr. Light has been a paid consultant for Astellas Pharma, Inc, Hepatares Therapeutics, NeuroSig, and Taketa Phamaceutical Company, Ltd, and received grants from Boerhinger-Ingelheim. Dr. Nuechterlein received grants from Janssen Pharmaceutica, Posit Science Corporation, and has been a paid consultant for Janssen Pharmaceutica, Astellas Pharma, Inc, Biogen, Inc., Genetech, Inc, MedinCell, Otsuka Pharmaceutical Co, Ltd, Takeda Pharmaceutical Company, Ltd, and Teva Pharmaceutical Industries, Ltd. No other disclosure is reported. "

3. Please ensure that you refer to Figure 1 in your text as, if accepted, production will need this reference to link the reader to the figure.

Reviewers' comments:

Reviewer's Responses to Questions

**Comments to the Author**

1. Is the manuscript technically sound, and do the data support the conclusions?

Reviewer #1: Yes

Reviewer #2: Yes

2. Has the statistical analysis been performed appropriately and rigorously? 

Reviewer #1: Yes

Reviewer #2: Yes

3. Have the authors made all data underlying the findings in their manuscript fully available?

Reviewer #1: Yes

Reviewer #2: Yes

4. Is the manuscript presented in an intelligible fashion and written in standard English?

Reviewer #1: Yes

Reviewer #2: Yes

5. Review Comments to the Author

Reviewer #1: In this study, the authors sought to determine whether age-related cognitive delcine is more evident in schizophrenia patinets compared to healthy control people. Data from a sizable number of subjects indicate that this is the case for performace on attention/vigilance. On the other hand, the age-dependent decline for working memory and verbal learning memory was, in fact, more apparent in control subjects.

Although some of these findings may not accord with the hypothesis raised, the persistent deterioration of cognitive function in schizophrenia, reportd here with the large number of participants, would add to the literature on the schizophrenia research.

Reviewer #2: Lee et al. investigated whether patients with schizophrenia show accelerated age-related decline in cognition using 1,415 schizophrenia patients and 1,062 healthy controls from the Consortium on the Genetics of Schizophrenia (COGS) Study. Multiple cognitive domains, the Letter-Number Span Task, two forms of the Continuous Performance Test, the California Verbal Learning Test, Second Edition, the Penn Emotion Identification Test and the Penn Facial Memory Test, were assessed. They found that patients showed greater age-related deterioration in some cognitive domains, while controls showed greater age-related deterioration in other domains. However, they suggested the age-related changes were much smaller than between case-control differences. This manuscript is well-written and straight-forwarded. The methods and results seem easy to understand. I recommend publishing this study with minor revisions.

There were some concerns as follows;

1. ‘Age-related changes (η2 of 0.001 to .008) were much smaller than between-group differences.’ Please add effect sizes for the group differences to compare effect sizes between the age-related changes and the group differences in abstract and discussion sections.

2. ‘Participants included 1,415 patients diagnosed with schizophrenia or schizoaffective disorder, depressed type’. Why did the authors recruit only depressed type of schizoaffective disorder? How many patients with schizoaffective disorder, depressed type, included in the patients? If the patients were restricted to only patients with schizophrenia, were their outcomes changed?

3. ‘Effect sizes are provided for any significant effects (i.e., partial eta square [η2 p]s.’ Please add scales for large, medium, and small effect sizes (η2).

6. PLOS authors have the option to publish the peer review history of their article (what does this mean?). If published, this will include your full peer review and any attached files.

Reviewer #1: No

Reviewer #2: Yes: Kazutaka Ohi, Department of Psychiatry and Psychotherapy, Gifu University Graduate School of Medicine, Gifu, Japan

---

## [Author Response · Author response to Decision Letter 0]

16 Apr 2020

We would like to thank the reviewers for their thoughtful comments and the Editor for the opportunity to submit a revised manuscript. As shown below, we address each comment from the Editor and Reviewer 2 and indicate where changes are made in the revised manuscript. The Reviewer 1 did not have any comments. The original comments from the Reviewer 2 are in italics. 

Editor: 

1. As requested, we ensure that our manuscript meets PLOS One’ style requirement, including those for file naming. 

2. As requested, we now clarify that the conflict of interests indicated in the Competing Interests section does not alter our adherence to all PLOS ONE policies on sharing data and materials (page XX). We also include our updated Competing Interests Statement in the cover letter.

3. As requested, we refer to Fig 1 in the revised manuscript (pages 10-12).

4. As requested, we now include captions for the Supporting Information Files at the end of the revised manuscript (page 17). 

Reviewer 2: 

1. ‘Age-related changes (n2 of .001 to .008) were much smaller than between-group differences.’ Please add effect sizes for the group differences to compare effect sizes between the age-related changes and the group differences in the abstract and discussion section. 

As requested, we now add effect sizes for the between-group differences (�2p of .005 to .037) in the abstract and discussion section (pages 10-12). 

2. ‘Participants included 1,415 patients diagnosed with schizophrenia or schizoaffective disorder, depressed type.’ Why did the authors recruit only depressed type of schizophrenia disorder? How many patients with schizoaffective disorder, depressed type, include in the patients? If the patients were restricted to only patients with schizophrenia, were their outcomes changed?

This is an important point and we appreciate the opportunity to clarify the rationale of our sample selection. We decided to recruit patients with schizophrenia or schizoaffective disorder, depressed type for the following reasons. First, findings from genetic studies suggest that schizoaffective disorder, depressed type, is more genetically similar to schizophrenia compared to schizoaffective disorder, bipolar type (Maj et al. 2001; Cardno et al., 2012). Second, schizoaffective disorder, depressed type, showed illness course similar to that of schizophrenia, compared to schizoaffective disorder, bipolar type (Andreasen et al., 1987; and Keshavan et al., 2011). This is now included in the revised manuscript (page xx). 

We also mention that our patient sample includes 59 patients with schizoaffective disorder, depressed type, and 1,315 patients with schizophrenia. Given the very small proportion of patients with schizoaffective disorder (about 4%), we decided to not run additional analyses excluding them. 

3. ‘Effect sizes are provided for any significant effects (i.e., partial eta square [n2p]s.’ Please add sales for large, medium, and small effect sizes (n2).

The general rule of thumb regarding the magnitude of effect size for partial eta square, is: 0.01 = small effect, 0.06 = medium effect, and 0.14 = large effect (Cohen, Milles, & Shevlin, 2001). We now mention this in the revised manuscript (page 9). 

References:

Andreasen, N. C., Rice, J., Endicott, J., Coryell, W., Grove, W. M., & Reich, T. (1987). Familial rates of affective disorder: A report from the National Institute of Mental Health Collaborative Study. Archives of General Psychiatry, 44, 461-469.

Cardno, A. G., Rijsdijk, F. V., West, R. M., et al. (2012). A twin study of schizoaffective-mania, schizoaffective-depression, and other psychotic syndromes. Am J Med Genet B Neuropsychiatr Genet, 159B, 172-182. 

Cohen, J., Milles, J., & Shevlin, M. (2001). Applying regression and correlations: A guide for students and researchers. London; Sage. 

Maj, M., Starace, F., & Pirozzi, R. (1991). A family study of DSM-III-R schizoaffective disorder, depressed type, compared with schizophrenia and psychotic and nonpsychotic major depression. Am J Psychiatry, 48, 612-616. 

Keshavan, M. S., Morris, D. W., Sweeney, J. A., Pearlson, G., Thaker, Gu, Seidman, L. J., Eack, S. M., & Tamminga, C. (2011). A dimensional approach to the psychosis spectrum between bipolar disorder and schizophrenia: The Schizo-Bipolar Scale. Schizophrenia Research, 133, 250-254.

---

## [Decision Letter · Decision Letter 1]

23 Apr 2020

The Effects of Age and Sex on Cognitive Impairment in Schizophrenia: Findings from the Consortium on the Genetics of Schizophrenia (COGS) Study

PONE-D-20-00548R1

Dear Dr. Lee,

We are pleased to inform you that your manuscript has been judged scientifically suitable for publication and will be formally accepted for publication once it complies with all outstanding technical requirements.

With kind regards,

Kenji Hashimoto, PhD

Section Editor

PLOS ONE

Additional Editor Comments (optional):

Reviewers' comments:

Reviewer's Responses to Questions

**Comments to the Author**

1. If the authors have adequately addressed your comments raised in a previous round of review and you feel that this manuscript is now acceptable for publication, you may indicate that here to bypass the “Comments to the Author” section, enter your conflict of interest statement in the “Confidential to Editor” section, and submit your "Accept" recommendation.

Reviewer #2: All comments have been addressed

2. Is the manuscript technically sound, and do the data support the conclusions?

Reviewer #2: Yes

3. Has the statistical analysis been performed appropriately and rigorously? 

Reviewer #2: Yes

4. Have the authors made all data underlying the findings in their manuscript fully available?

Reviewer #2: Yes

5. Is the manuscript presented in an intelligible fashion and written in standard English?

Reviewer #2: Yes

6. Review Comments to the Author

Reviewer #2: (No Response)

7. PLOS authors have the option to publish the peer review history of their article (what does this mean?). If published, this will include your full peer review and any attached files.

Reviewer #2: Yes: Kazutaka Ohi

---

## [Editor Report · Acceptance letter]

29 Apr 2020

PONE-D-20-00548R1 

The Effects of Age and Sex on Cognitive Impairment in Schizophrenia: Findings from the Consortium on the Genetics of Schizophrenia (COGS) Study 

Dear Dr. Lee:

I am pleased to inform you that your manuscript has been deemed suitable for publication in PLOS ONE. Congratulations! Your manuscript is now with our production department. 

With kind regards,

on behalf of

Prof. Kenji Hashimoto 

Section Editor

PLOS ONE